# Healthy Teleworking: Towards Personalized Exercise Recommendations

Maricarmen Almarcha [1,*], Natàlia Balagué [1] and Carlota Torrents [2]

1   Complex Systems in Sport Research Group, National Institute of Physical Education of Catalonia (INEFC), University of Barcelona (UB), Av. de l'Estadi 12-22, Anella Olímpica, E-08038 Barcelona, Spain; nbalague@gencat.cat
2   Complex Systems in Sport Research Group, National Institute of Physical Education of Catalonia (INEFC), University of Lleida, 1, 25192 Lleida, Spain; ctorrentsm@gencat.cat
*   Correspondence: mcalmarcha@gencat.cat

**Abstract:** Home-based teleworking, associated with sedentary behavior, may impair self-reported adult health status. Current exercise recommendations, based on universal recipes, may be insufficient or even misleading to promote healthy teleworking. From the Network Physiology of Exercise perspective, health is redefined as an adaptive emergent state, product of dynamic interactions among multiple levels (from genetic to social) that cannot be reduced to a few dimensions. Under such a perspective, fitness development is focused on enhancing the individual functional diversity potential, which is better achieved through varied and personalized exercise proposals. This paper discusses some myths related to ideal or unique recommendations, like the ideal exercise or posture, and the contribution of recent computer technologies and applications for prescribing exercise and assessing fitness. Highlighting the need for creating personalized working environments and strengthening the active contribution of users in the process, new recommendations related to teleworking posture, home exercise counselling, exercise monitoring and to the roles of healthcare and exercise professionals are proposed. Instead of exercise prescribers, professionals act as co-designers that help users to learn, co-adapt and adequately contextualize exercise in order to promote their somatic awareness, job satisfaction, productivity, work–life balance, wellbeing and health.

**Keywords:** exercise prescription; health; fitness; sedentary behavior; posture; affordances; somatic awareness; exergames

## 1. Introduction

Coronavirus (COVID-19) pandemic has produced a huge social and environmental impact on our lives, promoting organizational and behavioral changes with important implications for our lifestyle and our health status. One of the most outstanding is the proliferation of home-based teleworking, associated with sedentary behavior [1] and stress disorders [2].

The American College of Sports Medicine (ACSM) and The World Health Organization (WHO) recommend a physically active lifestyle, suggesting that adults participate in at least 150 to 300 min of moderate-intensity (3–6 METs), or 75 to 150 min of vigorous-intensity (>6 METs), or more for additional health benefits, of aerobic physical activity per week, together with strength exercises, to reduce the risk of chronic disease, including cardiovascular disease, type 2 diabetes mellitus, and certain types of cancer [3,4]. Such general exercise recommendations are addressed to healthy persons and clinical patients with multiple diseases and a wide age range. Following a one-size-fits-all approach, these recommendations have been recently questioned for: (a) reducing fitness to aerobic and strength exercises, (b) proposing a linear dose–response (exercise-benefits) relationship, and (c) their lack of personalization [5]. Systematic exercise prescription reviews reveal that even though most studies present favorable results applying the exercise programs

proposed by the aforementioned guidelines, there is a lack of evidence in several fields, a lack of high-quality studies [6] and a need for personalized recommendations [7].

Health and fitness concepts have been recently redefined based on a network physiology approach [8]. Sturmberg et al. [9] define health as a dynamic emergent state arising from nested networks interactions, and Pol et al. [10] define fitness as the capacity to survive in a broad range of contexts and point out that such capacity cannot be reduced to endurance and strength dimensions as proposed by current ACSM and WHO main guidelines. In fact, at a more general level, some authors have related fitness with the concept of intelligence, understood as the ability of systems to evade and escape states of reduced possibilities by creating functional compensatory synergies in dimensions other than those affected by reduced possibilities [11]. Under the framework of the Network Physiology of Exercise, it has been proposed that an adequately personalized exercise may promote the creation of functional synergies and healthy physiological network connectivity, characterized by functionality and flexibility, while a sedentary lifestyle may lead to dysfunctional, poor and weak connectivity [5].

In agreement with this new conception of physical fitness, the term exercise is used here in a wide sense. It includes all types of physical activities, not only those planned and structured pursuing the specific goal of improvement or maintenance of physical fitness [3]. Table 1 is summarizing and comparing some main characteristics of exercise recommended from an Exercise Physiology perspective, and from a Network Physiology of Exercise perspective.

**Table 1.** Comparison between exercise characteristics from an Exercise Physiology perspective and from a Network Physiology of Exercise perspective.

| Characteristics | Exercise Physiology | Network Physiology of Exercise |
|---|---|---|
| Main goal | Calorie expenditure Aerobic endurance and strength development | Functional diversity potential development |
| Recommendations | Universal | Personalized |
| Method | Programmed repetitions | Challenging variations |
| Dose/intensity | Preprogrammed | Contextually co-adapted |
| Practice | Monotonous, boring | Enjoyable, motivating |
| Monitoring | Based on technical devices | Based on somatic awareness |
| Professionals role | Prescribers | Co-designers |
| Users' role | Executers | Co-designers |

As health is a unique, individual adaptive state, the subjective dimension of health status plays a crucial role in a network understanding of health and disease [9]. In this sense, the subjective experience of health and illness (or poor health) can occur both in the absence and presence of objective disease. This distinction highlights the dynamic emergent nature of health and disease and indicates that in such a subjective scenario, the development of an adequate somatic awareness in individuals is a key element for safe exercise personalization (dose, type of exercise, etc.).

Scientific evidence has shown that regular exercise does improve not only the physical but also the psycho-emotional status, contributing to reduce negative emotion, relieve fatigue, improve sleep quality, and prevent cardiovascular and cerebrovascular diseases, among others [12]. Most of the studies testing the benefits of exercise on healthy people and clinical patients apply exercise protocols and standardized programs based on WHO and ACSM guidelines [3,13] and training recommendations [14]. Average intragroup values mask interindividual differences and neglect context-dependent variations within single individuals [15–17], and thus, cannot be applied to individualized exercise prescriptions [5].

In line with personalized exercise medicine, it has been proposed to reorient the main aims of exercise prescription, and accordingly, to redefine not only the roles of health care exercise professionals but also the user/patient role [12]. Particularly, it has been suggested to develop the autonomy of users/patients through their active involvement in the co-design of exercise proposals and the development of their somatic awareness. The selection of the type of activity, taking into account what is meaningful and attractive for the user, increases adherence to the practice, and periodization based on self-regulation and self-monitoring guarantees healthy and safe practices. As different internal and external constraints influence the individual mind-body states at very fast timescales [18], and these fast changes are hard to be captured through conventional monitoring systems, the development of somatic awareness [10,19] of users/patients is crucial. It may help to regulate and adjust, on a daily basis, active and resting periods, frequency, intensity, duration, etc., of exercise to promote healthy mind-body states.

We claim that education on self-regulation of psycho-emotional and physical states is essential to promote health and wellbeing during home-based teleworking. This requires that healthcare professionals guide the population from dependency to autonomy through the redefinition of fitness states, the aims and focus of the home-based exercise, co-adaptive and co-learning processes and the development of somatic awareness among users/patients. There is no universal way to reduce a sedentary lifestyle because there are huge interindividual differences, and personal contexts are continuously changing. New contexts suppose new challenges and new possibilities for learning and promoting creative behavior [20]. In this paper, we aim to propose, under the Network Physiology of Exercise approach, some personalized exercise medicine recommendations for keeping a healthy lifestyle under home-based teleworking conditions.

## 2. Home-Based Teleworking, Sedentary Lifestyle and Ideal Postures

Home-based teleworking is a form of work at home using information and communication technologies as support [21]. Working for prolonged periods with a computer, especially at home, is associated with a sedentary lifestyle, static and constraining posture, repetitive movements and extreme positions of the forearm and wrist [22].

Sedentary behavior includes all the activities that do not increase energy expenditure above resting levels, such as sleeping, sitting, lying down, and other forms of screen-based entertainment [23]. Such low levels of physical activity can have negative effects on health, wellbeing and quality of life [24]. Over the past five decades, jobs and occupations have increased the amount of seated technical work or desk-based office work [25]. A systematic review reveals that people with more active jobs had lower all-cause or cardiovascular disease mortality risk than those with jobs that involved mostly sitting [26]. A large-scale prospective cohort study in 220,000 Australians published an association between sitting and all-cause mortality across sexes, age groups, body mass index and physical activity levels [27]. Despite knowing how harmful it is to our health, we spend most of our daily time sitting. Offices, cinemas, cars, schools and restaurants are filled with chairs affording sitting. Standing, in contrast, is considered uncomfortable and even used in the past as a punishment (e.g., for children in schools).

Since the middle of the past century, changes in physical, economic, and social environments (transport, communication, workplace environment, and domestic entertainment technologies) have deeply changed our exercise habits. For such reason, moderate to vigorous exercise recommendations and physically active transportation has been recommended for the adult population [3]. One hour of vigorous exercise most days of the week, even if complying with the minimum of public health guidelines, cannot compensate fifteen hours of non-exercise awake time per day during which sitting is the predominant stance [28]. Sedentary comes from "sedere", which means "to sit". As the human body is made to be in frequent motion, sitting many hours a day, day after day, has widely recognized negative effects on health (reduced insulin sensitivity, impaired functioning of HDL cholesterol). This is why the program of sitting less and moving more has been recently promoted [29].

However, people sit because the places in which they spend their lives are structured around seats [30].

To compensate for the deleterious effects of sedentary habits in working places, some solutions such as ergonomic office and workstation design have emerged. They are intended to contribute to weight control/loss through additional energy expenditure, relief and prevent musculoskeletal pain (acute and chronic), and improve cardiometabolic health (e.g., adjustable sit-stand desktops). Workplace interventions that promote standing breaks and sit-stand adjustable workstations show improvements in health markers and increase work productivity, efficiency and collaboration among employees [27,31,32]. Although stand-up working at the computer doubles the energy expenditure over the 8 h workday and is a good alternative [33], the focus of healthy computer working has been put on preventing spine-related health problems derived from erroneous sitting postures. However, the questions are: Is there an ideal ergonomic position and sitting posture? Is energy expenditure the main health problem?

*The Ideal Sitting Posture*

The ideal sitting posture has been widely debated concerning back pain problems. A common belief is that spinal pain is caused by sitting, standing, or bending "incorrectly." Similarly, sitting up straight or looking for pelvis head alignment has been considered a healthy posture to prevent backache [34]. However, research has shown that there is no relationship between the shape and curves of the back with pain [35], neither movement screening for the prevention of pain in the workplace. Moreover, the complete consensus among experts concerning which is the best sitting posture does not exist because of disagreements on what constitutes a neutral spine posture and what is the best sitting posture. Even the chronic ideal posture for a long time can create as many problems as sitting all day [36], and also, repetitive movements can lead to injuries [37]. Despite the absence of strong evidence to support these common beliefs, health interventions and ergonomic assessments in offices are prescribed to get a "correct" posture and prevent pain.

The biomechanical definition of posture as a static configuration of the body in the space does not explain individual spinal variability (e.g., shapes, sizes), which is an adaptable structure capable of safely moving and loading in a variety of postures. From an enactive point of view, posture is embodied and dynamic: the action emerges from the interaction of emotions, intentions, motivations while the action is still ongoing [38]. Therefore, the change of posture has a wider dimension that includes personal constraints (e.g., muscular characteristics, height and proportions, psychological state) [39].

In general, posture studies highlight the importance of personalized management, as pain is influenced by numerous biopsychosocial factors. Claus et al. [40] proposed that any sustained sitting posture could result in fatigue, discomfort and pain, including the "bad" or the "good" postures if they persist uninterrupted for extended periods. This statement suggests that postural variability or regular movement can be beneficial in reducing maintained sitting posture risks. In this line, in the last decades, dynamic sitting approaches have been proposed, considering that subjects with back pain assume more static and sustained postures while sitting. Dynamic sitting is referred to the use of chairs or equipment that facilitates an increased trunk motion and spinal micro-movements, such as stability balls, chairs with a degree of motion or passive motion devices on the seat [41]. Comfortable postures vary between individuals, so it is useful to encourage people to move and explore different postures while sitting. The main problem is not the posture itself but the amount of time spent keeping it without changes. Hence, instead of looking for an ideal general posture, individuals and work companies should promote environments where movement and variety of postures are required and encouraged.

## 3. Current Recommendations for Maintaining Fitness Levels at Home during the COVID-19 Lockdown

Despite limited space or lack of special equipment, the WHO and the ACSM recommended 150 to 300 min of moderate-intensity or 75 to 150 min of vigorous-intensity PA

per week during the COVID-19 lockdown [3]. The US Department of Health and Human Services, Office of Disease Prevention and Health Promotion, recently added a webpage entitled "Staying Active While Social Distancing", providing support for physical activity and guidance [29].

For desk-based workers, the recommendations proposed progressions from 2 h/day to 4 h/day of standing and light activity. To achieve this stage, they recommended regularly breaking up seated-based work with standing-based work, the use of sit-stand desks, or the taking of short active standing breaks. Interrupting prolonged sitting time has proved to have metabolic health benefits [42]. Regular interruptions of 3-min of light–intensity activities for 20 min of sedentary time have been suggested [43]. WHO exemplifies some practical home-based exercises and muscle-strengthening exercises available at their website (e.g., squats, planks, bridges, chair dips, etc.) to be performed for 10–15 repetitions up to five times with 1-min rest between sets to maintain PA during coronavirus mobility restrictions [3]. In addition, to increase exercise motivation, they propose Internet-delivered interventions that one can follow through electronic devices as a tool for everyone [44].

The use of ergometers and other technical devices has also been recommended to increase fitness levels and, particularly, total energy expenditure [45]. The purchase of ergometers, usually available in fitness centers where healthcare and exercise professionals usually refer their clients and patients, has become very popular during confinement, with a 170% rise in the purchase of sports equipment [46]. Despite their possibilities to provide vigorous-intensity activities, indirect measures of energy expenditure, and a precise quantification and regulation of resistance and cadence of movements, such devices present some limitations concerning similar activities performed under open-air conditions. In contrast, open-air activities provide a dynamical environment with exploration opportunities and movement variability [47]. Furthermore, there are other crucial health and fitness-related issues besides energy expenditure, often underestimated in the recommendations, that in interaction with exercise, have a relevant role on psychobiological states (i.e., nutrition, alcohol, smoking, stress or quality of life).

## 4. Alternatives to Current Recommendations

### 4.1. Improving Environmental Affordances

Affordance-responsiveness is a central feature of the everyday skillful activity of humans [48]. Affordances are possibilities for action provided by the environment, including possibilities for social interaction [49]. The notion of affordances are studied in different areas such as philosophy/phenomenology [50,51], sports/ecological psychology [52–54], affective science [55], and neuroscience [56]. However, affordances are not only possibilities of action; they invite behavior. For example, a room full of chairs affords sitting, an open space invites a conversation, and an extended hand invites a handshake. When a person encounters a meaningful affordance, a state of bodily action readiness occurs [57], and this is why chairs can "suck us in". If we radically change the affordances available in a certain environment, behavioral changes will occur [58]. Hence, the challenge lies in transforming both the physical and the social environment to reduce long-lasting immobility during home-based teleworking.

Architects and artists have manipulated work environments to create new affordances [59]. The newly created landscapes aim to develop behavioral changes through carefully selected and meticulously designed interventions in urban or rural areas that set the desired developments in motion. The multidisciplinary studio Rietveld Architecture-Art-Affordances (RAAAF) and visual artist Barbara Visser built an enactive art installation called The End of Sitting, a landscape without chairs that integrates many affordances for standing and increases bodily activity and wellbeing [59]. These designs lead to a range of affordances and offer users the freedom that characterizes everyday unreflective action [60]. Such actions emerge from the individual-environment interaction [10], and new affordances may create new movement habits during the working time [58]. For example, new landscapes without chairs, like the installations proposed by The End of

Sitting project, lead to more freedom of movement and new posture and body activity habits of participants during working periods associated with wellbeing [59].

### 4.2. Exergaming Approach

Virtual reality is an environment generated by computer technology, which allows user interaction and creates in the user the feeling of being immersed in it [61]. The interface environment provides full exploration and movement variability in people with disabilities [62]. Game properties (use of rewards, goals, feedback) and game engagement make virtual reality games use a great option for therapeutical purposes, motor skills learning, individualized learning and socialization [63]. Although games developed for rehabilitation purposes are expensive and hardly accessible, another type of game has been developed: exergames.

Exergaming is based on technology that tracks and projects body movements into an avatar on screen. Variety in exercise options provides a field for customizing games based on user's needs and motivations, such as walking, dancing, yoga, swimming, tennis, boxing or golf [44]. In current training interventions, prescribed exercises are executed with supervision in real time, but in exergaming at home, there is no real-time feedback or instruction. This sometimes requires a co-designing process between users/patients and professionals to adjust the exercise intervention according to the individual's response to exercise, progress and corresponding fitness and therapeutic needs [64].

Due to these advantages, exergames have been integrated into prevention and rehabilitation programs in different pathological conditions [65–69] turning out to be more motivating and engaging than conventional rehabilitation programs for children with obesity [70], healthy older adults [71,72], patients with Parkinson's disease [73], acquired brain injury [74], ataxia [75] and multiple sclerosis [76]. They have even reduced anxiety and stress levels during the isolation period in the COVID-19 pandemic [77]. The integration of exergames seems to have a positive effect on adherence, and thus, is potentially beneficial for the long-term effectiveness of rehabilitation programs [78]. To facilitate access to validated exergames for end-users and healthcare institutions, digital libraries can be found at https://openrehab.org (accessed on 31 December 2020) and https://seriousgames-portal.org (accessed on 31 December 2020).

Use of artificial intelligence to customize exercise prescriptions based on psychobiological factors could be a complementary tool to influence exercise behavior and movement habits. For example, the computerized exercise expert system (CEES) customizes exercise prescriptions based on personal questions administered to patients [79]. Performance evaluation of a recommending interface (PERI) offers the possibility to adjust and personalize exercise recommendations according to an evaluation performed from the deep learning neural network approach [80].

### 4.3. Mobile Applications with Fitness Purposes

Fitness products based on mobile applications have become popular due to the varied and safe home exercise options they offer. They do not depend on specific gym equipment (e.g., treadmill, bike) and may offer virtual free lessons through sports clubs and fitness instructors. Some of these apps also provide a virtual community and data tracking with the latest wearable technology, such as Apple watches, Garmin devices, or Fitbits smartwatches. Health metrics feedback transferred to healthcare professionals can supply valuable information to develop and readjust user's exercise characteristics. Nevertheless, relying on external devices for exercise monitoring may limit the development of somatic awareness, which is a necessary ingredient for developing adequate self-monitoring abilities and contribute to the user's autonomy and self-regulation.

### 5. New Perspectives and Practical Recommendations

Healthy home-based teleworking does not only consist of following the available general recommendations and guidelines of exercise prescription for healthy and clinical

populations. There are huge differences in the daily physical activity of teleworkers in their previous experiences, injuries, diseases, preferred activities, etc. Accordingly, personalized healthy teleworking requires taking into account the following practical recommendations:

-   *The acknowledgment of risks associated with prolonged immobility*

    According to current guidelines, there is an extended belief that 30–40 min of moderate exercise per day is enough to keep fitness and healthy habits in adults. However, the effects of limited-time exercise bouts cannot compensate for the deleterious effects of lying down for the rest of the day. There is also a traditional assumption that there is an ideal sitting posture while working. Nevertheless, the immobility associated with keeping the same posture during long periods can induce, more than avoiding, health problems. Movement is necessary to coordinate organs [12], activate psychobiological functions and stimulate the body and the mind to keep the psycho-emotional and physical states that ensure job satisfaction, productivity, work–life balance, wellbeing and health.

-   *Redefining health and fitness objectives*

    It is recommended that healthcare professionals, exercise professionals and users/patients familiarize themselves with the recently redefined concepts of fitness and health [9,10], and reorient fitness objectives accordingly. Health states vary with personal and environmental constraints, and fitness objectives should adapt to it. Due to its multidimensional nature, fitness cannot be only developed through few conditional training dimensions (endurance and strength) [5,9,10]. Instead, it should focus on developing the functional diversity potential in a wide, multidimensional and personalized way. This means varying and adapting not only exercise challenges but also finding compensatory synergies through other abilities (e.g., intellectual, artistic) in order to evade and escape states of reduced possibilities.

-   *Co-designing and co-adapting personalized exercise programs*

    As there are no identical personal and environmental constraints and there is no universal way to develop fitness, healthcare professionals, exercise professionals and users are encouraged to collaborate in creating and developing personalized exercise programs. Based on flexible criteria, these programs pursue to promote adherence and adapting exercise periodization on a daily basis, according to the immediate constraints and affordances. Users' self-regulation of daily activity is the final aim of program interventions, and the role of professionals, instead of just prescribing exercise, is guiding users/patients from dependency to autonomy [5].

-   *Development of user's somatic awareness*

    Varied exercise and movement experiences may contribute to increase the functional diversity potential of users and develop their fitness [11]. Diverse and non-repetitive activities promote the creation of new synergies that may improve connectivity among organs and physiological systems. In turn, movement diversity provides rich body-mind information that enhances somatic sensitivity and awareness. The attention towards body signs may enhance self-monitoring abilities and develop further the somatic and informed awareness. It is then recommended avoiding or limiting the use of gadgets and applications for exercise and workout prescription and adjustments. For instance, it is preferable to take breaks or active pauses during work based on a subjective feeling of uneasiness or discomfort rather than on preprogrammed alarms.

-   *Creation of personalized working environments limiting sitting and affording exercise and posture variations*

    Home teleworkers are often able to organize their time and space according to individual needs. Exploration and discovery of personalized exercise and movement possibilities is a challenging process that requires users to be aware of the huge amount of available possibilities for changing postures and exercising. Exergaming contributes to diversifying home-based personalized exercise programs offering possibilities for exercising

individually, in family, regulating intensity, and selecting motivating activities according to personal preferences.

The use of adjustable working stations allows changing posture, standing up or sitting, and provides movement possibilities and variations during the day. Employees indicate that some tasks can be completed standing or exercising at moderate intensities, e.g., checking emails and making phone calls. However, other tasks such as reading or writing, involving greater concentration, are better performed during sitting. According to some authors, the commitment to standing is influenced by the perception of improved productivity and experience of health benefits [81]. Enabling floor to work provides further movement possibilities and openness to new affordances. This means more readiness to engage with relevant opportunities for action in a concrete situation [58]. The exploration of new movement possibilities (e.g., local, global, micro-movements, different types of contractions, different muscle group activation, changing postures, making active pauses, stretching) can also be encouraged through personal challenges.

- *Complementary proposals*

Sitting less and moving more are recommended to change the intensity of some daily activities (e.g., reducing car use, running instead of walking, dancing, walking during phone calls, shopping using bikes). Ergometers can also be used while watching videos, chatting, or listening to audio. It is important to prioritize outdoor activity whenever possible. Research from a variety of scientific fields suggests that physical activity in nature enhances health-related quality of life and long-term adherence to physical activity [82].

## 6. Limitations and Future Lines of Research

Proposing exercise criteria and workout self-regulation based on subjective monitoring and somatic awareness, instead of on prescribed exercise recipes, may allow an adequate contextualization and personalization of physical activity during teleworking, but may increase potential risks associated with exercise (e.g., injuries, overtraining). However, there is a lack of research confirming the prevalence of risks related to self-prescribed or unsupervised exercise [83], in contrast to self-prescribed medication [84]. To avoid potential risks, it is recommended developing somatic awareness at an early age, and promote it during the intervention process. It is relevant to highlight here that an excessive or obsessive focus on body awareness may intensify the symptomatology of users predisposed to suffering hypochondria. In this sense, more research related to the development of somatic awareness and, in particular, to the less used concept of informed awareness [84]. The term informed awareness refers to the information about oneself (e.g., proprioception, interoception) in relation to the environmental information.

It is worth pointing out that systematic, repetitive exercise, as proposed by current guideline recommendations, is one more option among all exercise possibilities that can be offered. It may be as well contextually valid, especially when user's preferences are addressed to it. As health and fitness have a subjective dimension, it seems recommendable to respect users' preferences and develop progressively their understanding related to the advantages derived from varied exercise modalities and challenges.

Finally, although computer technologies and mobile applications may suppose useful and easy solutions for exercise prescription and fitness evaluation, it is suitable that users do not simply rely on technical devices for selecting exercise options and assessing their fitness and health status. The continuous use of external devices replacing important human abilities related to survival, such as somatic and informed awareness, can be detrimental at long term [85]. In addition, the development of self-responsibility towards own fitness and health is of key importance [84].

## 7. Conclusions

Home teleworking, associated with sedentary behavior, may suppose a threat to health. Recommended standardized exercise programs, based on the one-size-fits-all approach, maybe insufficient and even misleading to promote fitness and health. Twenty

to thirty min of moderate exercise cannot compensate for the deleterious effects of sitting the rest of the day, as might be misinterpreted from current guidelines. A personalized exercise counseling should place particular attention on (a) educating professionals and users on the redefined concepts and objectives of health and fitness, (b) design interventions focused on developing the functional diversity potential of users, (c) promote the creation of functional compensatory synergies in multiple dimensions to evade user's states of reduced possibilities, (d) co-design and co-adapt exercise interventions together with users, (e) contribute to develop user's somatic awareness and workout self-regulation competencies, (f) creating adequate environmental contexts affording movement variation possibilities and motivating exertion, instead of focusing on ideal and unique solutions (e.g., the ideal posture).

A better understanding of health and fitness objectives by professionals and users is crucial to develop adequate person-centered exercise solutions and counteract sedentary tendencies during teleworking. The active participation of users in the co-design and co-adaptation of personalized exercise programs is of key importance for exercise adherence and health prevention. Computer technologies may provide support to the co-designed programs but cannot replace users' decisions and users' monitoring abilities based on somatic awareness. It is crucial that healthcare and exercise professionals bear in mind that one of the main aims of the intervention process is developing user's autonomy and self-responsibility towards one's own fitness and health. Empirical research is warranted to study the effects of cooperative and self-prescribed exercise on producing safe and effective interventions regulated by somatic awareness.

**Author Contributions:** Conceptualization, M.A. and N.B.; original draft preparation, M.A. and N.B.; writing—review and editing, M.A., N.B. and C.T.; visualization, M.A.; supervision, N.B.; funding acquisition, M.A. and N.B. All authors have read and agreed to the published version of the manuscript.

**Funding:** This work was supported by the National Institute of Physical Education of Catalonia (INEFC), Generalitat de Catalunya. M.C.A. is supported by the project "Towards an embodied and trans-disciplinary education" granted by the Ministerio de Educación y Formación Profesional of the Spanish government (FPU19/05693).

**Institutional Review Board Statement:** Not applicable.

**Informed Consent Statement:** Not applicable.

**Data Availability Statement:** Not applicable.

**Conflicts of Interest:** The authors declare no conflict of interest.

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
