# Peer review of "Healthy Teleworking: Towards Personalized Exercise Recommendations"

_sustainability, doi:10.3390/su13063192_

Round 1

Reviewer 1 Report

This is an interesting study regarding healthy teleworking. The focus of this paper is the human body movement. This theory work as an organized set of authors ideas. However, sometimes there is a mix between physical activity and Exercise. The authors may well clarify booth and differentiate during the manuscript.

The authors present some issues regarding working from home. It is interesting. However, using PC, smartphone, etc., may require most of the time. How about to better explore the gadgets use to promote physical activity? Such as apps, agendas or reminders to perform specific physical activities exercises? Otherwise, workers may forget or due working concentration.

In some specific points of chapter 4 is hard to understand the practical applications, the aims of the information and novelty.

More specific commentaries below:

Introduction: please reinforce that self-prescribed exercise may be an health risk; whereas, physical exercise professionals prefer to prescribe physical activity. The introduction is manly based on physical exercise. This reviewer believes that, considering the impossibility of some people to have professionals prescribing exercise it would be better physical activity.

L109 and 110: merge paragraphs.

L189: miss parentheses.

  1. It would be interesting for the authors explore the physical activity promoting apps to improve workers health.

L.304: this is an hard objective to reach. That is because it is hard to have general programs to be individualize without precise evaluation. Why not focus on physical activity?

L311. Misses a practical application.

Readjust conclusion based on the commentaries.

Reviewer 2 Report

The Authors have provided an abundant listof references, providing a good background to their paper. The paper consists of review and proposals regarding healthy teleworking.

It is necessary to develop the Abstract to provide apart from the goal also the main considerations and suggestions of the paper.

Since the paper can be seen as a foresight in the field of personalized exercise recommendations and healthy teleworking, in Section 4 I would expect an extra Subsection with references to sources in order  to provide a short review of mobile applications which allow personalized exercise recommendations.

The new subsection to mention novel artificial intelligence frameworks which facilitate recommendations such as deep learning framework which can be used for quasi-intelligent recommending of exercise in a similar way as in paper titled Deep learning-enhanced framework for performance evaluation of a recommending Interface with varied recommendation position and intensity based on eye-tracking equipment data processing.  Another interesting reference which should be used here is paper titled Development of an exercise expert system for older adults.

The limitations of the study and directions of future research need to be covered in the Conclusions, and contributions were clearly outlined.

Please consider the recommendations as friendly, constructive suggestions to improve the already very good quality of your manuscript.  

With best regards!

Reviewer 3 Report

Thanks you for this interested manuscript. I consider that this information could be highly useful for the scientific community.

Abstract: If it not a limit of words, I would included here more about your acknowledges (about the “key points” of the healthy home teleworking, at the end of your paper)

Introduction:

  • included the new ACSM recommendations-reference (2020)
  • Line 83: torrents et al. (in press), please, delete this reference or included it as Vancouver reference
  • I recommend included before the hypothesis a brief paragraph speaking about the increasing of teleworking because the COVID-19 and some references about it

Home based-telework, sedentarism, immobility and the ideal posture

  • When you speak about vigorous/moderate intensity, I recommend included definition of these terms.
  • The 31 reference is missing in the text (line 125 = 30, line 127 = 132)
  • Might could be interested included some reference to Physical Activity Breaks (a methodology to increase PA in students that they are sometimes 5-6 hours in class without any physical activity)

Current recommendations for maintaining fitness levels at home during confinement

  • A bracket is missing (line 170)
  • I recommend included WHO recommendations 2020 (not 2010, line 168)
  • Line 171 separate 2 of from
  • Line 189-193 please include some reference

Alternatives to current recommendations

  • Included or before golf (line 229)
  • Separate [68] (line 236)

Other concerns

  • Delete “achnowledgements sections”
  • Included the dates from journals references with blond format
  • Reference 31 is missing of the date and the title is repeated
  • References: included the page to the different references that are missing (8, 9, 10…)

From my point of view, it could be interesting if you make a table with an example of the recommendations with the different points, for instance, with the five point do you write in the “new perspectives and practical recommendations” section. But, with some example for a “typical person”, explaining: possible risks associates; what should be their fitness purpose; an example of how to design (or simply design) a personal physical activity program; 1-2 tips for develop the users somatic awareness; 3-4 tips from creation of personalized working environments

I hope this recommendations will be useful for you and thanks you for this paper.

Round 2

Reviewer 1 Report

The authors have improved the manuscript. However, there are severall paragraphs missing references. For example:

Section 4.3;

L.378-383;

L.384-393: There's only one reference. That's oberved on the paragraphs below. Review manuscript may have a considerable scientific support.

Please insert more references.

Author Response

The authors have not found the references mentioned by the reviewer. 

L.378-383; 

Correspond to the conclusion section

L.384-393: There's only one reference. That's oberved on the paragraphs below. Review manuscript may have a considerable scientific support.

There is no reference because it is the table 1 and funding section

It could have been a confusion when referencing the lines.